# Microaxial Left Ventricular Assist Device in Cardiogenic Shock: A Systematic Review and Meta-Analysis

**DOI:** 10.3390/life12101629

**Published:** 2022-10-18

**Authors:** Shien Ru Tan, Christopher Jer Wei Low, Wei Lin Ng, Ryan Ruiyang Ling, Chuen Seng Tan, Shir Lynn Lim, Robin Cherian, Weiqin Lin, Kiran Shekar, Saikat Mitra, Graeme MacLaren, Kollengode Ramanathan

**Affiliations:** 1Yong Loo Lin School of Medicine, National University of Singapore, Singapore 119228, Singapore; 2Saw Swee Hock School of Public Health, National University of Singapore, Singapore 119228, Singapore; 3Department of Cardiology, National University Heart Centre, Singapore 119228, Singapore; 4Adult Intensive Care Services, The Prince Charles Hospital, Brisbane, QLD 4032, Australia; 5Institute of Health and Biomedical Innovation, Queensland University of Technology, Brisbane, QLD 4000, Australia; 6Faculty of Medicine, University of Queensland, Brisbane, QLD 4072, Australia; 7Faculty of Medicine, Bond University, Gold Coast, QLD 4226, Australia; 8Intensive Care Unit, Dandenong and Casey Hospital, Monash Health, Melbourne, VIC 3175, Australia; 9Cardiothoracic Intensive Care Unit, National University Heart Centre, National University Health System, Singapore 119228, Singapore

**Keywords:** Impella, cardiogenic shock, ventricular assist device, meta-analysis

## Abstract

Microaxial left ventricular assist devices (LVAD) are increasingly used to support patients with cardiogenic shock; however, outcome results are limited to single-center studies, registry data and select reviews. We conducted a systematic review and meta-analysis, searching three databases for relevant studies reporting on microaxial LVAD use in adults with cardiogenic shock. We conducted a random-effects meta-analysis (DerSimonian and Laird) based on short-term mortality (primary outcome), long-term mortality and device complications (secondary outcomes). We assessed the risk of bias and certainty of evidence using the Joanna Briggs Institute and the GRADE approaches, respectively. A total of 63 observational studies (3896 patients), 6 propensity-score matched (PSM) studies and 2 randomized controlled trials (RCTs) were included (384 patients). The pooled short-term mortality from observational studies was 46.5% (95%-CI: 42.7–50.3%); this was 48.9% (95%-CI: 43.8–54.1%) amongst PSM studies and RCTs. The pooled mortality at 90 days, 6 months and 1 year was 41.8%, 51.1% and 54.3%, respectively. Hemolysis and access-site bleeding were the most common complications, each with a pooled incidence of around 20%. The reported mortality rate of microaxial LVADs was not significantly lower than extracorporeal membrane oxygenation (ECMO) or intra-aortic balloon pumps (IABP). Current evidence does not suggest any mortality benefit when compared to ECMO or IABP.

## 1. Introduction

The incidence of cardiogenic shock (CS) has increased in recent years, yet long-term mortality has not substantially improved in the last 20 years [1]. It is associated with significant multi-organ failure and in-hospital mortality reaching in excess of 60% [2,3]. Amongst survivors, up to 20% are re-admitted within 30 days [1]. Acute myocardial infarction is the most common cause of CS and accounts for 10% of patients with CS [1,4]. A spectrum of disease exists in cardiogenic shock—the Society of Cardiovascular Angiography and Interventions (SCAI) classifies CS from Stages A (at-risk) to E (in extremis). Within Stage C (classic) CS, patients typically present with hypoperfusion requiring either inotropes or temporary circulatory support devices [5]. Various temporary circulatory support devices are available for these patients—this ranges from counterpulsation devices such as the intra-aortic balloon pump (IABP), percutaneously inserted left ventricular assist devices (pLVADs, including microaxial and centrifugal), paracorporeal VADs and extracorporeal membrane oxygenation [6,7]. 

Despite the wide range of options for temporary circulatory support, outcomes remain variable [7,8,9,10,11,12,13]. Microaxial LVADs placed retrogradely across the aortic or pulmonary valves [7,14] are increasingly being used to support patients with cardiogenic shock of various etiologies [14,15,16,17,18], or as a “bridge to decision” in end-stage heart failure [19]. They reduce the ventricular afterload and ventricular end-diastolic pressure, and increase the mean arterial pressure [20] and cardiac output [21]. Smaller percutaneous devices have been authorized for use for up to 4 days, [22] whereas larger surgically inserted devices are authorized for use for up to 14 days [23]. 

In addition to the multiple device options, existing studies and reviews investigating its use report favorable survival outcomes and safety outcomes in patients with CS [9,24]. However, the outcomes of microaxial LVADs based on the various types and different etiologies of CS have not been elucidated in detail [24]. In addition, potential predictors of mortality have yet to be explored. We conducted this systematic review and meta-analysis to investigate the short- and long-term mortality outcomes and device-related complications of microaxial LVADs in all etiologies of CS, and to explore the potential risk factors associated with mortality. 

## 2. Methodology

### 2.1. Search Strategy and Selection Criteria

This review was registered on PROSPERO (CRD42020202807) and conducted in accordance with the Preferred Reporting Items for Systematic Reviews and Meta-Analysis (PRISMA) statement (Appendix A) [25]. We searched MEDLINE, Embase and Scopus databases from 1 January 2003 to 13 July 2022 using the keywords ‘Impella’ and ‘cardiogenic shock’ (Appendix A). We included studies published in English, reporting on ≥10 non-overlapping adult patients (>18 years) receiving microaxial LVADs for CS. In cases of overlapping patient data, we included the larger study. We excluded studies reporting on animals and where device was inserted prophylactically or electively during percutaneous coronary intervention. We also excluded those studies where outcomes were not stratified by device option in CS, and national or international registry databases that could contribute to duplication of patient data. 

### 2.2. Data Extraction

Data collection included study design (author and study name, year of publication, country, setting, number of patients), patient demographics (age, gender, comorbidities), pre-LVAD clinical characteristics (body mass index, left ventricular ejection fraction [LVEF], comorbidities), etiology of CS (acute myocardial infarction cardiogenic shock [AMICS] or non-myocardial infarction cardiogenic shock [NMICS]), device characteristics (mode of insertion, cannulation access, concomitant extracorporeal membrane oxygenation [ECMO] use, duration of support) and outcomes of interest (in-hospital mortality, 30 days, 90 days, 6 months, 1 year and device-related complications).

### 2.3. Risk of Bias and Certainty Assessment

Risk of bias in individual studies was assessed using the appropriate Joanna Briggs Institute (JBI) checklists. Egger’s test was used to assess the possibility of publication bias. As inter-study heterogeneity can be misleadingly large when assessed using I2 statistics for observational studies [26], we used the Grading of Recommendations, Assessment, Development, and Evaluations (GRADE) approach to rate the certainty of evidence [27,28]. The screening of articles, data collection and risk of bias assessment were conducted independently by two reviewers (TSR and NWL), and any conflicts were resolved by a third reviewer (KR).

### 2.4. Outcomes

The primary outcome was short-term mortality, defined as 30-day mortality or in-hospital mortality, whichever was longer. Secondary outcomes include long-term mortality at 90 days, 6 months and 1 year, and device-related complications (device malfunction, access-site bleeding, hemolysis, limb ischemia and stroke). Appendix A summarize the definitions of CS, device-related complications and severity of LVEF [29].

### 2.5. Statistical Analysis

For continuous variables, we pooled the means and standard deviations (SDs) in accordance with Wan et al. [30]. Categorical data are reported as pooled proportions with 95% confidence intervals (CIs), whereas continuous outcomes are reported as pooled means with 95% CIs. All analyses were conducted in R4.0.1. Random effects meta-analyses (DerSimonian and Laird) were conducted using the Freeman–Tukey double arcsine transformation, and 95% CIs were computed using the Clopper–Pearson method [31,32,33]. We pooled the results of the propensity-score matched (PSM) studies and RCTs together as previous studies have shown that the estimates obtained from PSM studies are similar and as robust as RCTs [34,35,36]. Sensitivity analysis was conducted by excluding studies with higher risks of bias (defined as <7). Planned subgroup analyses were conducted with continuity correction of 0.5 to allow for inclusion of studies with zero events, and included the geographical region (Europe, North America or Asia), etiology of CS (AMICS or NMICS), the mode of insertion (percutaneous (which comprises Impella 2.5 and CP) or surgical (which comprises Impella 5.0 and Impella 5.5)), cannulation access for insertion (axillary or femoral), duration of support (more or less than 4 days), concomitant use of ECMO, IABP prior to microaxial LVAD use and pre-LVAD LVEF (above or below 20%). Summary-level meta-regression was conducted if there was a minimum of 6 data points in order to explore sources of heterogeneity and to identify potential prognostically relevant study-level covariates [37].

### 2.6. Role of the Funding Source

This study had no funding source.

## 3. Results

From 4173 articles, we reviewed 206 full-text articles. In total, we included 71 studies (63 observational, 8 PSM/RCTs) detailing 4280 adult patients that reported on the use of microaxial LVADs in CS (Figure 1) [16,38,39,40,41,42,43,44,45,46,47,48,49,50,51,52,53,54,55,56,57,58,59,60,61,62,63,64,65,66,67,68,69,70,71,72,73,74,75,76,77,78,79,80,81,82,83,84,85,86,87,88,89,90,91,92,93,94,95,96,97,98,99,100,101,102,103,104,105,106]. The findings of the one-armed observational studies and the findings of the PSM/RCTs are reported separately. Of the observational studies, 34 were reported by centers from Europe, 25 from North America, 3 from Asia and 1 from South America, whereas all of the RCTs and PSMs were reported by centers from Europe. Percutaneously inserted devices were more commonly used than surgically inserted devices.

### 3.1. Demographics of Included Studies

Table 1 presents the baseline demographics of the observational studies. Among the 63 studies, 14 studies reported on patients with AMICS, 4 studies reported on patients with NMICS, and 38 studies reported on both patients with NMICS and patients with AMICS. The etiology of cardiogenic shock was not reported in 8 studies. Patients were predominantly male (75.0%, 95%-CI: 70.9% to 78.8%), and were supported for an average of 6.2 days (95%-CI: 4.7 to 7.7). The pooled intensive care unit (ICU) stay was 13.7 days (95%-CI: 10.2 to 17.2), and the pooled hospital length of stay was 20.6 days (95%-CI: 13.0 to 28.2).

Table 1 refers to pooled demographics of 384 patients across the PSM studies and RCTs. The pooled age was 62.2 years, and the majority (82.2%) were male. The pooled duration of microaxial LVAD support was 2.9 days, and the pooled ICU stay was 8.6 days. The pooled hospital stay was 15.3 days. 

### 3.2. Primary Meta-Analysis

Amongst the observational studies (63 studies, 3896 patients), the pooled short-term mortality was 46.5% (95%-CI: 42.7% to 50.3%, Figure 2). As all studies had a JBI score of ≥7, sensitivity analysis excluding studies with higher risks of bias was not possible. We excluded studies with a JBI score of <10 as an exploratory analysis, and this yielded similar pooled estimates for the short-term mortality (44.7%, 95%-CI: 40.2% to 49.2%).

There were no significant differences in short-term mortality with respect to the etiology of CS, mode of insertion and concomitant use of ECMO. Patients who presented with AMICS (52.1%, 95%-CI: 46.8% to 57.3%, 14 studies) had a comparatively higher short-term mortality than patients who presented with NMICS (42.0%, 95%-CI: 33.5 to 50.8%, 5 studies, *p* = 0.085). Mortality was significantly higher among patients receiving concomitant ECMO (51.5%, 95%-CI: 47.1% to 55.9%, 8 studies) than patients receiving microaxial LVADs only (44.6%, 95%-CI: 39.6% to 49.6%, 40 studies, *p* = 0.043). 

No significant differences were found in short-term mortality when considering the geographical location (North America, South America, Europe or Asia), patient demographic factors (pre-LVAD LVEF (≤20% or >20%)) or device factors (duration of microaxial LVAD support (≤4 days or >4 days) and cannulation site (axillary or femoral)). Appendix A summarizes the results of the subgroup analysis. 

Univariate meta-regression found significant associations between mortality and previous cerebrovascular accidents (regression coefficient (B): 0.29, 95%-CI: 0.15 to 0.56, *p* = 0.038) and hyperlipidemia (B: 0.68, 95%-CI: 0.02 to 0.32, *p* = 0.030), and an inverse association with the duration of the device support (B: −0.015, 95%-CI: −0.015, 95%-CI: −0.022 to −0.009, *p* < 0.0001). However, there was no significant association between mortality and patient demographics, including age, other comorbidities (hypertension, diabetes mellitus, previous acute myocardial infarction, heart failure, smoking) and pre-LVAD LVEF. Appendix A summarizes the meta-regression analysis.

Among the PSM studies and RCTs, the pooled short-term mortality (Figure 3) was 48.9% (95%-CI: 43.8% to 54.1%) From one study that compared microaxial LVADs to IABP alone, microaxial LVADs did not significantly reduce the risk of mortality (RR: 0.94, 95%-CI: 0.58–1.53, *p* = 0.81). Five studies provided a comparison between microaxial LVADs and other devices; we report these findings qualitatively. From two studies comparing microaxial LVADs with IABP, one study found approximately 46% of patients expired in both cohorts [87], and similar findings were reported in the other (hazard ratio for mortality: 0.96, *p* = 0.92 [77]. One study found that microaxial LVADs (49.4%) were associated with a trend to lower mortality compared to ECMO (61.4%, *p* = 0.16) [64], which was echoed by another PSM study (55% vs. 67.5%, *p* = 0.36) [105], while another study found that concurrent microaxial LVAD with ECMO (47%) significantly reduced mortality compared to ECMO alone (80%, *p* < 0.001) [79]. Finally, microaxial LVAD was shown to improve mortality (20%) compared to patients without any mechanical circulatory support (47%, *p* = 0.0024) [88].

### 3.3. Secondary Outcomes

#### 3.3.1. Long-Term Mortality

The pooled 90-day, 6-month and 1-year mortality was 41.8% (95%-CI: 34.4% to 49.3%, 5 studies, 448 patients), 51.1% (95%-CI: 45.2% to 57.0%, 9 studies, 676 patients) and 54.3% (95%-CI: 48.9% to 59.7%, 10 studies, 881 patients), respectively, amongst the observational studies (Figure 4). Two PSM studies reported a mortality at the 6-month follow up that ranged between 36.6% (12/33) and 75% (45/60) [58,74]. One PSM study reported a 1-year mortality rate of 60%.

#### 3.3.2. Complications

Table 2 shows the top five device-related complications (hemolysis, access-site bleeding device malfunction, limb ischemia, stroke) reported amongst 50 observational studies (3101 patients). Hemolysis (24.9%, 95%-CI: 14.9% to 36.4%, 1708 patients, 23 studies), access-site bleeding (25.8%, 95%-CI: 14.7% to 38.5%, 1679 patients, 23 studies) and device malfunction (6.0%, 95%-CI: 3.1% to 9.3%, 690 patients, 17 studies) were the three most common complications in this patient cohort. From the PSM studies and RCTs, the pooled incidence of stroke (three studies) was 0.4% (95%-CI: 0.0% to 2.4%), whereas hemolysis occurred in 40.8% (95%-CI: 4.4% to 84.4%, four studies) of patients. Bleeding was reported among four studies (6.4%, 95%-CI: 3.3% to 10.4%), and three studies reported on device malfunction (3.2%, 95%-CI: 0.0% to 26.9%). Finally, 6.8% (95-CI: 0.0% to 21.6%) of patients (six studies) suffered from limb ischemia.

### 3.4. Risk of Bias and Certainty of Evidence Assessment

Using appropriate JBI checklists, all studies were of high quality (score of ≥7, Appendix A). We assessed the certainty of evidence for all primary and secondary outcome measures using the GRADE approach (Appendix A). For both observational studies and RCTs, the certainty of evidence was high according to the GRADE evaluation for our primary outcome of short-term mortality and that of long-term mortality, whereas the complications were deemed to be of moderate-to-high certainty. Egger’s test found that P_egger_ was 0.96, indicating that publication bias is unlikely.

## 4. Discussion

This review comprising 71 studies and 4280 patients demonstrated that microaxial LVAD in CS was associated with mortality rates approaching 50%. Patients were predominantly middle-aged males. The 90-day, 6-month and 1-year mortality (observational studies) was 41.8%, 51.1% and 54.3%, respectively. Short-term mortality was relatively higher in patients with surgical insertion compared to percutaneous insertion. Comorbitidies including previous cerebrovascular accidents and hyperlipidemia were associated with mortality, whereas longer durations of device support were associated with survival. 

Our study provides further insights into the characteristics of microaxial LVAD devices that may affect mortality. We found that patients receiving surgically inserted devices had a relatively higher mortality rate than percutaneously inserted devices. This is likely to be because multifactorial-percutaneously inserted devices generate a maximum of 2.5 to 4.0 L/min of blood flow, [107] whereas surgically inserted devices generate up to 5.0 and 6.0 L/min [107,108]. Patients with more severe cardiogenic shock may have higher support requirements and intrinsically higher mortality rates due to their clinical presentation. In addition, the surgical insertion of devices might increase the rates of surgical site infection and bleeding. Finally, higher flows generated by surgically inserted devices may lead to higher rates of hemolysis. We also found that the duration of the device support was not associated with a higher mortality. This is contrary to previous studies that have shown that the use of microaxial LVADs for >4 days led to an increased mortality and duration of hospital and coronary care unit stay [109]. Nonetheless, this could be attributed to immortal time bias, which has been described in observational studies [110] and in patients on life-saving devices [111,112], where patients in the treated group have to survive and be event-free until the treatment definition is fulfilled [113]. 

Mortality rates for CS remain high despite timely goal-directed medical management [7,114,115]. The variable survival rates of CS between the use of mechanical cardiac support devices is evident from the IABP-SHOCK I and II trials that showed that IABP did not significantly improve 30-day survival [10,116], whereas the international Extracorporeal Life Support Organization registry found that 42% of patients receiving venoarterial ECMO survived to discharge [117]. This contrasts with the 51% survival rate of patients receiving microaxial LVADs in the United States [18]. Similarly, in our observational cohort of patients with microaxial LVAD support, we observed short-term survival rates of 53%. However, survival rates of 70% have been reported in advanced cardiac centers with robust protocols comprising the stringent selection criteria team-based management of CS [118,119]. The concomitant use of microaxial LVADs and ECMO is an area of increasing interest to improve outcomes. Microaxial LVADs unload the left ventricle (LV) and may help offset the LV distension secondary to retrograde aortic blood flow in patients on peripheral venoarterial ECMO [120]. Our study found that patients receiving concomitant ECMO had a significantly higher mortality rate than those receiving microaxial LVADs alone (51.5% vs. 44.6%, *p* = 0.04). However, this can be confounded by the severity of cardiogenic shock, and VA-ECMO may only be initiated in the context of cardiogenic shock refractory to other therapies. As such, it is unclear whether VA-ECMO causes an increase in mortality, or if it is simply initiated in patients with more severe cardiogenic shock. 

The long-term mortality reported in our review was higher compared to those reported in major trials on microaxial LVADs [17,121]. The reasons may be multifactorial: both RCTs had fewer patients with a smaller range of etiologies of CS, and robust patient selection criteria and management protocols. On the other hand, patients recruited in the observational studies were heterogenous in selection and management. The higher incidence of complication rates could also have impacted the long-term outcomes. The most frequently reported device-related complication was hemolysis, which was higher than those reported in previous registry reviews [24,122,123]. There was also a discrepancy between RCTs and PSM studies, and observational studies in the incidence of hemolysis (40.8% vs. 23.8%) and access site bleeding (6.4% vs. 25.8%). Possible reasons include a longer pooled duration of device support in observational studies compared to RCTs, varying definitions of hemolysis and the predominant use of percutaneous devices with a smaller pump design. Access-site bleeding was reported in 15 studies with a pooled prevalence of 19.4%, similar to the USpella cohort [122] but lower than the EUROSHOCK cohort [108,123,124]. Notably, our study found that the pooled incidence of limb ischemia was comparable between the observational studies and RCTs (6.3% vs. 8.2%), and was lower compared to ECMO and IABP, whereas the incidence of access-site bleeding was higher compared to ECMO and IABP [37,125,126]. Nonetheless, the incidence of limb ischemia and bleeding may have been affected by multiple factors, such as the use of anticoagulants or presence of peripheral vascular disease, for which, adequate data were not clearly available [7]. 

The strengths of this review include a comprehensive search strategy and robust inclusion criteria that encompassed all etiologies of CS and types of devices used. It also included a detailed analysis of various patient and intervention factors and their impact on mortality outcomes. Nonetheless, we recognize several limitations. First, there is significant heterogeneity in patient demographics, definitions, variations in patient selection, practices and reporting patterns and the observational nature of the included studies, which we tried to account for by using subgroup and meta-regression analyses. Meta-regression analyses are also inherently constrained by a lack of power, resulting in an increased risk of type II errors. Almost all of the analyses have also been limited to North America and Europe, whereas studies from Asia remain scarce. Hence, the results might not be generalizable to other parts of the world where healthcare systems and workflows are different. Nonetheless, our subgroup analysis on geographical location did not find any significant difference in short-term mortality. Moreover, the GRADE assessment suggested a high certainty in the evidence for the primary outcome and long-term mortality, whereas complications were of moderate to high certainty. With scores of 7 or higher, JBI critical appraisal also deemed all 71 articles as high quality and suitable for inclusion.

## 5. Conclusions

This review summarizes the mortality outcomes and complications of microaxial LVADs in patients with CS. Short-term mortality was 46.5% whereas 6-month and 1-year mortalities were 51% and 54%, respectively. Complications such as hemolysis and access site bleeding were high as reported in the observational studies. Nonetheless, the use of temporary circulatory support in cardiogenic shock remains inherently challenging as patients are usually critically ill with multi-organ pathologies, and patient care is heterogenous. In addition, the current evidence base is limited in concluding whether or not microaxial LVADs confer a survival benefit in patients with CS. Further RCTs are warranted to better assess the effectiveness and role of microaxial LVADs in CS.

## Figures and Tables

**Figure 1 life-12-01629-f001:**
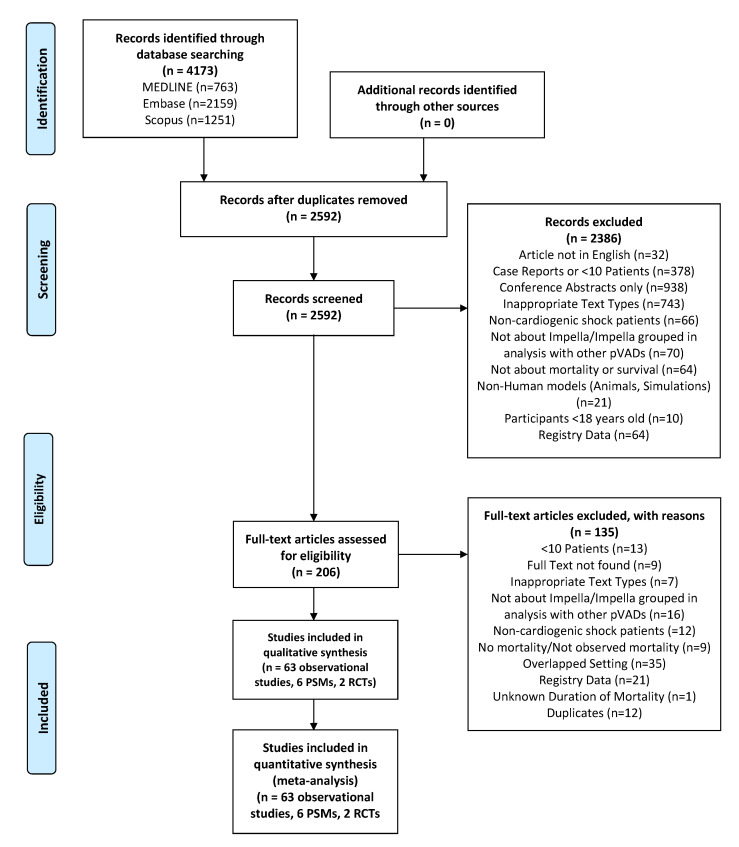
Flow diagram of selection of articles based on PRISMA statement. Abbreviations: PRISMA = Preferred Reporting Items for Systematic Reviews and Meta-Analyses; pVAD = percutaneous ventricular assist device; RCT = randomized controlled trial.

**Figure 2 life-12-01629-f002:**
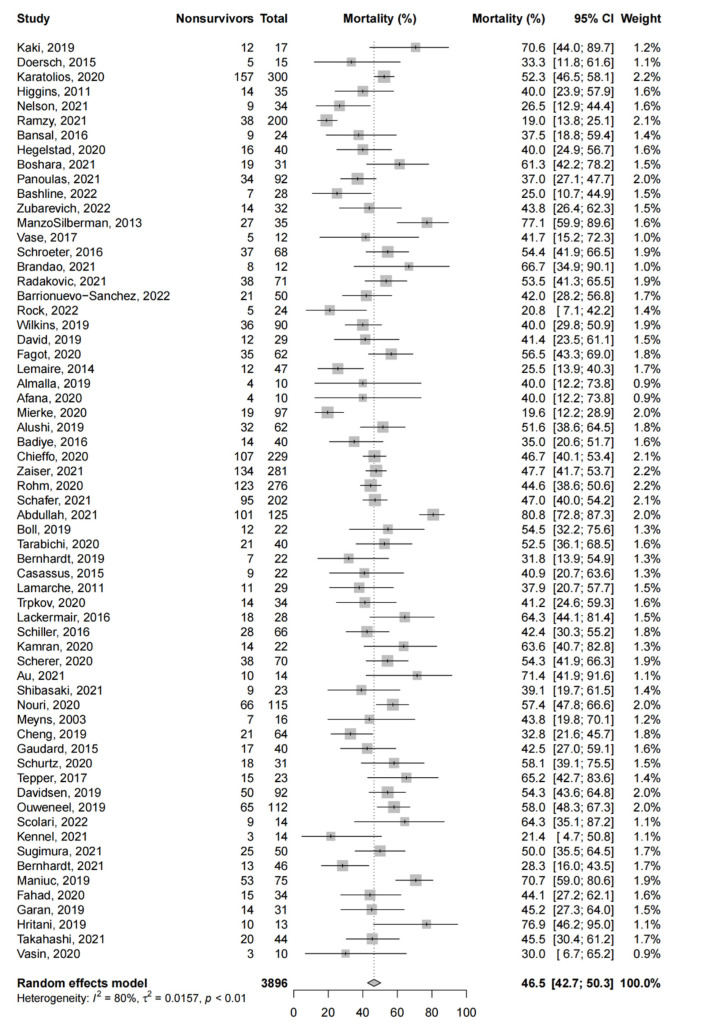
Short-term mortality in observational studies. Forest plot summarizing the short-term mortality of patients receiving microaxial LVAD for cardiogenic shock amongst observational studies. Abbreviations: CI = confidence interval.

**Figure 3 life-12-01629-f003:**
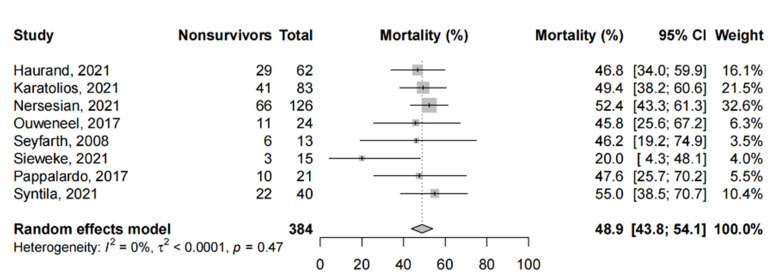
Short-term mortality in propensity-score matched studies and randomized controlled trials. Forest plot summarizing the short-term mortality of patients receiving microaxial LVAD for cardiogenic shock amongst propensity-score matched studies and randomized controlled trials. Abbreviations: CI = confidence interval.

**Figure 4 life-12-01629-f004:**
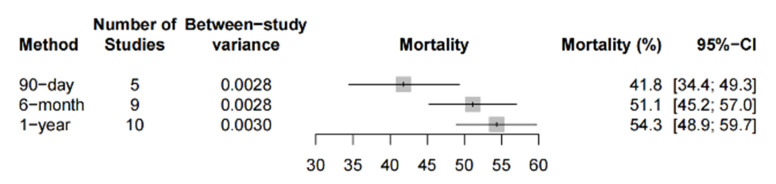
Long-term mortality in observational studies. Forest plot summarizing the 90-day, 6-month and 1-year mortality of patients receiving microaxial LVAD for cardiogenic shock based on observational studies. Abbreviations: CI = confidence interval.

**Table 1 life-12-01629-t001:** Demographics of included studies.

Demographics	Study Type	Studies	Pooled Estimate	95%CI
**Patient Demographics**				
Age (years)	Observational	59	61.6	60.1–63.0
	RCT/PSM	8	62.2	58.8–65.6
Male (%)	Observational	59	75.0%	70.9–78.8%
	RCT/PSM	8	82.2%	70.5–90.0%
**Pre-Impella Characteristics**				
Body Mass Index (kg/m^2^)	Observational	18	27.4	26.8–28.0
	RCT/PSM	2	25.7	23.1–28.2
LVEF (%)	Observational	32	25.3	22.8–27.9
	RCT/PSM	3	33.8	31.8–35.8
Hypertension (%)	Observational	41	53.5%	45.7–61.3%
	RCT/PSM	6	55.6%	38.1–71.8%
Hyperlipidemia (%)	Observational	34	35.4%	25.2–46.4%
	RCT/PSM	2	35.1%	10.9–70.7%
Diabetes Mellitus (%)	Observational	48	30.6%	27.2–34.1%
	RCT/PSM	7	28.8%	20.1–39.4%
Smoking (%)	Observational	40	26.8%	18.6–35.8%
	RCT/PSM	3	46.2%	33.2–59.7%
Heart Failure (%)	Observational	24	20.3%	10.0–32.7%
	RCT/PSM	Was not reported in any of the studies		
Previous AMI (%)	Observational	26	20.7%	14.3–27.9%
	RCT/PSM	5	32.5%	15.6–55.5%
Cerebrovascular Accident (%)	Observational	29	7.5%	5.0–10.2%
	RCT/PSM	3	1.3%	0.4–4.0%
**Post-device Characteristics**				
Duration of device support (days)	Observational	38	6.2	4.7–7.7
	RCT/PSM	7	2.9	1.5–4.3
Length of ICU stay (days)	Observational	13	13.7	10.2–17.2
	RCT/PSM	2	8.6	6.5–10.8
Length of hospital stay (days)	Observational	6	20.6	13.0–28.2
	RCT/PSM	3	15.3	12.0–18.7

Abbreviations: AMI: acute myocardial infarction, AMICS: Acute myocardial infarction cardiogenic shock, CI: confidence interval, ICU: intensive care unit, LVEF: left ventricular ejection fraction, PSM: propensity-score matched study, RCT: randomized controlled trial.

**Table 2 life-12-01629-t002:** Complications of Observational Studies.

Complication	Studies	Pooled Proportion (%)	95% CI	I^2^
Hemolysis	32	24.9	14.9 to 36.4	94.8%
Access-site Bleeding	23	25.8	14.7 to 38.5	94.9%
Device Malfunction	17	5.9	3.1 to 9.3	58.8%
Limb Ischemia	32	6.1	3.7 to 8.9	76.1%
Stroke	32	5.5	2.9 to 8.5	77.6%

## Data Availability

No new data were created or analyzed in this study. Data sharing is not applicable to this article.

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
