# Peer review of "Microaxial Left Ventricular Assist Device in Cardiogenic Shock: A Systematic Review and Meta-Analysis"

_life, 2022, doi:10.3390/life12101629_

Round 1

Reviewer 1 Report

This manuscript is a systematic review and meta-analysis summarising the use of microaxial left ventricular assist devices (LVADs) for cardiogenic shock.

It is also very important to note that the reported mortality rates for microaxial LVAD were not significantly lower than those for extracorporeal membrane oxygenation (ECMO) or intra-aortic balloon pump (IABP).

This result indicates that there is no suggested benefit for mortality when compared to ECMO or IABP.

The flow diagram (figure1) is blurred and the text is not clear and difficult to read. Please correct the text and graphs to make them easier to read. 

Author Response

This manuscript is a systematic review and meta-analysis summarising the use of microaxial left ventricular assist devices (LVADs) for cardiogenic shock.

It is also very important to note that the reported mortality rates for microaxial LVAD were not significantly lower than those for extracorporeal membrane oxygenation (ECMO) or intra-aortic balloon pump (IABP).

This result indicates that there is no suggested benefit for mortality when compared to ECMO or IABP.

Response: Thank you.

The flow diagram (figure1) is blurred and the text is not clear and difficult to read. Please correct the text and graphs to make them easier to read. 

Response: Thank you. Reviewer #2 had also made a similar comment regarding this issue. We initially submitted our manuscript without the template used by the production team. It is possible that during the processing of the manuscript, the resolution might have decreased. For this revision we have reuploaded figure 1 directly in the manuscript, hopefully when this is returned to the reviewers the quality of the figure will be improved. We also attach a copy of Figure 1 with our responses to the comments. We will work with the production team to ensure that Figure 1 is published with sufficient resolution.

Reviewer 2 Report

Tan et al. performed a systematic review on the use of microaxial left ventricular assist devices in cardiogenic shock patients. They conclude from their analyses that the current evidence does not suggest any mortality benefit when compared to ECMO or IABP, which is an important information for clinical practice. The search strategy is appropriate; and the inclusion/exclusion criteria are well defined.

The authors should consider introducing the different assist devices in the introduction for non-expert readers. This might broaden the audience for the manuscript.

Minor comments:

The list of authors seems to be incomplete.

Citations should be included systematically in the sentences not after.

Resolution of Figure 1 needs improvement.

Line 170: Please remove the statement that mortality was higher if it was not significant.

Line 242: Please remove the “..”

References: Please correct the style according to the Journals instructions.

The use of color and highlighting in the Suppl. is not explained and seems not to be justified. 

Author Response

Tan et al. performed a systematic review on the use of microaxial left ventricular assist devices in cardiogenic shock patients. They conclude from their analyses that the current evidence does not suggest any mortality benefit when compared to ECMO or IABP, which is an important information for clinical practice. The search strategy is appropriate; and the inclusion/exclusion criteria are well defined.

Response: Thank you.

The authors should consider introducing the different assist devices in the introduction for non-expert readers. This might broaden the audience for the manuscript.

Response: Thank you. We have added some exposition on the classification of cardiogenic shock (CS) and its heterogenous spectrum, as well as the range of temporary circulatory devices which can be used in cardiogenic shock. We attach the relevant section below:

A spectrum of disease exists in cardiogenic shock – the Society of Cardiovascular Angiography and Interventions (SCAI) classifies CS from Stages A (at-risk) to E (in extremis). Within Stage C (classic) CS, patients typically present with hypoperfusion requiring either inotropes or temporary circulatory support devices(5). Various temporary circulatory support devices are available for these patients – this ranges from counterpulsation devices such as the intra-aortic balloon pump (IABP), percutaneously inserted left ventricular assist devices (pLVADs, including microaxial and centrfigual), and extracorporeal membrane oxygenation(6,7).

Minor comments:

The list of authors seems to be incomplete.

Response: Thank you. We had initially submitted our manuscript without the template used by the production team. In the process of converting the manuscript to the MDPI-appropriate format, it is possible that the authors list was moved next to the list of affiliations. We have gone through this and amended it in the revision 

Citations should be included systematically in the sentences not after.

Response: Thank you. We have made changes to the manuscript to reflect this accordingly.

Resolution of Figure 1 needs improvement.

Response: Thank you. As per the first minor comment, we initially submitted our manuscript without the template used by the production team. It is possible that during the processing of the manuscript, the resolution might have decreased. For this revision we have reuploaded figure 1 directly in the manuscript, hopefully when this is returned to the reviewers the quality of the figure will be improved. We have also uploaded a copy of the Figure with our response, please find attached. We will work with the production team to ensure that Figure 1 is published with sufficient resolution.

Line 170: Please remove the statement that mortality was higher if it was not significant.

Response: Thank you, we have removed this.

Line 242: Please remove the “..”

Response: Thank you, we have removed this.

References: Please correct the style according to the Journals instructions.

Response: Thank you. We understand that the Journal allows for references in any style, provided that a consistent formatting is used throughout. We have gone through the references to ensure that they adhere to the Journal’s instructions, and formatted them accordingly as well.

The use of color and highlighting in the Suppl. is not explained and seems not to be justified.

Response: Thank you. We apologise for the oversight, and have removed the colour and highlighting 

Reviewer 3 Report

Dear Authors,

I think that your paper is well designed. I have a question that you should also clarify in the text: do you attribute the higher mortality to the surgical insertion of the assist device or to the intrinsic severity of the cardiogenic shock? or to both maybe? You should be more precise about this point.

Best

Author Response

Dear Authors,

I think that your paper is well designed.

Response: Thank you.

I have a question that you should also clarify in the text: do you attribute the higher mortality to the surgical insertion of the assist device or to the intrinsic severity of the cardiogenic shock? or to both maybe? You should be more precise about this point.

Response: Thank you. We interpret this comment in relation to Reviewer #1, who has asked for the sentence on the difference in mortality between percutaneous and surgical insertion of microaxial LVADs to be removed in view of the lack of significance.

As the reviewer points out, patients with more severe cardiogenic shock will require higher degrees of support, in which case surgically inserted microaxial LVADs might be used over percutaneously inserted microaxial LVADs. It is difficult to directly attribute higher mortality rates in surgically inserted microaxial LVADs to either the complications of surgical insertion, or the intrinsic severity of cardiogenic shock. Rather, it is likely that these higher mortality rates are multifactorial: patients with increasing support requirements are less likely to survive, the mode of insertion may increase the rates of bleeding and infection, and the higher blood flows generated may increase the rates of hemolysis. We have added this in the discussion section to further clarify the relative difference in mortality rates. This is summarised below:

Our study provides further insights into the characteristics of microaxial LVAD devices which may affect mortality. We found that patients receiving surgically inserted devices had a relatively higher mortality rate than percutaneously inserted devices. This is likely multifactorial - percutaneously inserted devices generate a maximum of 2.5 to 4.0L/min of blood flow,(108) while surgically inserted devices which generate up to 5.0 and 6.0L/min,(108,109). Patients with more severe cardiogenic shock may have higher support requirements, and intrinsically higher mortality rates due to their clinical presentation. In addition, the surgical insertion of devices might increase the rates of surgical site infection and bleeding. Finally, higher flows generated by surgically inserted devices may lead to higher rates of hemolysis.